# Analysis of the Phytochemical Composition of Pomegranate Fruit Juices, Peels and Kernels: A Comparative Study on Four Cultivars Grown in Southern Italy

**DOI:** 10.3390/plants10112521

**Published:** 2021-11-19

**Authors:** Anna Montefusco, Miriana Durante, Danilo Migoni, Monica De Caroli, Riadh Ilahy, Zoltán Pék, Lajos Helyes, Francesco Paolo Fanizzi, Giovanni Mita, Gabriella Piro, Marcello Salvatore Lenucci

**Affiliations:** 1Dipartimento di Scienze e Tecnologie Biologiche ed Ambientali (Di.S.Te.B.A.), Università del Salento, Via Prov.le Lecce Monteroni, 73100 Lecce, Italy; anna.montefusco@unisalento.it (A.M.); danilo.migoni@unisalento.it (D.M.); monica.decaroli@unisalento.it (M.D.C.); fp.fanizzi@unisalento.it (F.P.F.); gabriella.piro@unisalento.it (G.P.); 2Istituto di Scienze delle Produzioni Alimentari (ISPA)-CNR, Via Prov.le Lecce-Monteroni, 73100 Lecce, Italy; miriana.durante@ispa.cnr.it (M.D.); giovanni.mita@ispa.cnr.it (G.M.); 3Laboratory of Horticulture, National Agricultural Research Institute of Tunisia (INRAT), University of Carthage, Ariana 1040, Tunisia; bn.riadh@gmail.com; 4Horticultural Institute, Hungarian University of Agriculture and Life Sciences, 2100 Gödöllő, Hungary; pek.zoltan@uni-mate.hu (Z.P.); Helyes.Lajos@uni-mate.hu (L.H.)

**Keywords:** agri-food by-products, antioxidants, ascorbic acid, flavonoids, fruit peels, juices, phenolic compounds, *Punica granatum* L., seeds, sugars

## Abstract

The increasing popularity of pomegranate (*Punica granatum* L.), driven by the awareness of its nutraceutical properties and excellent environmental adaptability, is promoting a global expansion of its production area. This investigation reports the variability in the weight, moisture, pH, total soluble solids, carbohydrates, organic acids, phenolic compounds, fatty acids, antioxidant activities, and element composition of different fruit parts (juices, peels, and kernels) from four (Ako, Emek, Kamel, and Wonderful One) of the most widely cultivated Israeli pomegranate varieties in Salento (South Italy). To the best of our knowledge, this is the first systematic characterization of different fruit parts from pomegranate cultivars grown simultaneously in the same orchard and subjected to identical agronomic and environmental conditions. Significant genotype-dependent variability was observed for many of the investigated parameters, though without any correlation among fruit parts. The levels of phenols, flavonoids, anthocyanins, and ascorbic and dehydroascorbic acids of all samples were higher than the literature-reported data, as was the antioxidant activity. This is likely due to positive interactions among genotypes, the environment, and good agricultural practices. This study also confirms that pomegranate kernels and peels are, respectively, rich sources of punicic acid and phenols together, with several other bioactive molecules. However, the variability in their levels emphasizes the need for further research to better exploit their agro-industrial potential and thereby increase juice-production chain sustainability. This study will help to assist breeders and growers to respond to consumer and industrial preferences and encourage the development of biorefinery strategies for the utilization of pomegranate by-products as nutraceuticals or value-added ingredients for custom-tailored supplemented foods.

## 1. Introduction

Pomegranate (*Punica granatum* L.) belongs to the olygotypic Punicaceae (Bercht. & J. Presl) family, are related to the Lythraceae, but distinguished by most authors based on some morphological features. These include, mostly, the fusion of the ovary with the receptacle and the characteristic structure of the fruit, known as balausta (Figure 1). Botanically, the balausta is a multilocular multiseeds fruit, which develops from an inferior syncarpous ovary. Its pericarp, comprising the reddish exocarp (rind) and the external part of the mesocarp (albedo), is tough and leathery. The inner spongy mesocarp also forms yellowish, papery, plate-like infoldings (locular septa) covering the individual groups of tightly packed seeds, improperly known as arils. The endocarp consists of two rows of multi-ovule chambers (locules) arranged asymmetrically (usually 2–3 in the upper row and 6–9 in the bottom one). The calyx is persistent and shaped like a crown. Kernels, the inner parts of the arils, comprise the embryo protected by the tegmen and the sclerotic mesotesta and are irregularly arranged on the placenta and enclosed in a fleshy and juicy coat (sarcotesta) [1,2]. The natural growth habit of the plant is to form deciduous shrubs or small trees, producing at maturity (5 years old) up to 24 kg of fresh fruits per tree [3]. Generally, the rind plus the albedo (peels) constitutes 30–40% of the fruit’s fresh weight (fw), while the remaining edible part, which constitutes, on average, about 52% of the total fruit’s weight, consists of 78% juice and 22% kernels, with a certain variability among the over-500 cultivated varieties [4]. Rind and aril color, juice sweetness, acidity, and astringency, as well as kernel hardness, are the main factors affecting consumer preference [5].

Pomegranate is native to Transcaucasia and Central Asia, from modern-day Iran and Turkmenistan to northern India, but has been cultivated and naturalized since ancient times throughout the Mediterranean region and Eastern Asia. Today, pomegranate is widely cultivated throughout the Middle East and Caucasus region, the semi-arid areas of Southeast Asia and Africa, Latin America, the Southeast USA, and the Southern states of Europe, including Spain, Italy, and Greece [6,7]. In the Boreal Hemisphere, pomegranate fruits are typically in season from September to February, while in the Austral Hemisphere the production spans from March to July. 

The pomegranate fruit has been embraced by many cultures throughout the world, who have always highly appreciated the fruit for its health-promoting attributes. In many ancient occidental and oriental cultures, pomegranate fruit was granted enormous symbolic and apotropaic value, being considered the emblem of eternity and resurrection [8]. 

Currently, the pomegranate fruit is renowned and marketed as a “superfood”. Its numerous bioactive components act, in fact, as antioxidant and anti-inflammatory molecules, and represent an effective preventive tool against cancer, heart disease, childhood cerebral ischemia, Alzheimer’s disease, diabetes, arthritis, obesity, male infertility, bacterial infections, and radiation-induced tissue damage. Furthermore, extracts from all its parts, including arils, peels, and kernels, have demonstrated therapeutic properties [4,7,8,9]. Recently, a commercial pomegranate seed extract has been proposed as a source of antioxidants to develop an innovative synbiotic formulation to reduce dysbiosis and uremic toxins in patients with chronic kidney disease [10]. Although pomegranate peels and kernels have been traditionally valorized as animal feed, most often, large-scale juice processing industries manage them as waste with considerable hauling and disposal costs [11,12]. However, several biorefinery approaches have been proposed to process these agri-food by-products into added-value bio-based commodities and functional ingredients for food, medicinal, and cosmetic use, creating profitable revenues along the supply chain [13,14]. Pomegranate peel is, in fact, a huge source of phenolics, flavonoids, ellagitannins, hydroxybenzoic acids, and many other bioactive molecules, and exhibits higher antioxidant activity than the edible portion [4,15,16]. Likewise, kernels are a rich reserve of vitamins, proteins, minerals, fibers and, above all, fatty acids with beneficial effects on health [17,18]. Pomegranate kernel oil, as well as peel extracts, also showed estrogenic activity due to the presence of non-steroidal phytoestrogens [19].

Until 2010, large-scale pomegranate cultivation in Italy was almost non-existent. Production was limited to a few tens of hectares or plants scattered in parks and public or private gardens. Currently, over 1500 hectares are cultivated, especially in the south. Sicily, Puglia, Calabria, Campania, and Lazio are the regions most affected by this rapid expansion. In the last 5 years, pomegranate fruit consumption has seen a surge higher than 25%. A similar trend was registered at the European level and worldwide. Currently, Italy ranks 16th in export and 12th in import of fresh pomegranates, with a share of 1.97% and 2.15%, respectively. The thriving market for fresh product is further expanded by the growing interest of Italian customers in pomegranate-based processed products, including juices, purees, natural colorings, fresh-cut fruit, wines, vinegars, beauty products, and food supplements. In Salento (Apulia, Italy), the spread of pomegranate is further encouraged, as it represents a valid alternative crop to the decimation of olive trees determined by *Xylella fastidiosa* infections.

The chemical composition of the fruits differs depending on the genotype, growing region, environmental conditions, ripening stage, cultivation practices, and storage conditions [4,20,21]. Thus, the aim of this work was to evaluate a few attributes related to fruit quality, as well as the content of some classes of bioactive molecules (phenolics, flavonoids, anthocyanins, ascorbic acids (AsA) and dehydroascorbic acids (DAsA)), and 25 chemical elements of the juice, peels, and kernels of four different Israeli pomegranate cultivars (Ako, Emek, Kamel, and Wonderful one) grown in Southern Italy. We also estimated the hydrophilic (HAA), lipophilic (LAA), and total (TAA) antioxidant activities of the different fruit parts, as well as the fatty acid profiles of kernel oils. This study confirms that genotype significantly affects the chemical composition and, consequently, the antioxidant activity of ripe pomegranate fruits, both the edible portion (arils or juice) and the industrial processing by-products (peels and kernels). Furthermore, it emphasizes the need to evaluate pomegranate biodiversity to improve fruit nutritional value, as well as the convenience to identify novel strategies for the recovery of valuable chemicals from their industrial by-products with the perspective of enabling pathways to agricultural sustainability.

## 2. Results and Discussion

### 2.1. Pomological and Fruit Quality Attributes 

Figure 2 reports the external and internal appearance of the ripe fruits of the four investigated Israeli pomegranate genotypes, the average fruit weight, and the relative proportion (weight percentage) of the isolated juice, kernel, and peel fruit parts. 

Fruit weight differed significantly among cultivars. Wonderful One had the highest average fruit weight (506 g), followed by Ako (438 g), Kamel (347 g), and Emek (329 g) (Figure 2a). A similar ranking was obtained for fruits harvested from the same orchard in the years 2012 and 2013; however, compared to our findings, the average weight was higher for Wonderful One (622.3 g) and slightly lower for Ako (322.9 g) and Emek (309.8 g) [22]. Much lower values were reported for Wonderful One (324–481 g) and Ako (151–350 g) cultivars grown in different Mediterranean countries, with a marked variability over the years [3]. A similar average weight (522.7 g) was, instead, found in Wonderful One pomegranate fruits grown in a commercial orchard in the north of Apulia (Foggia, Italy), with a gradual decrease in reduced irrigation regimes (down to 332.5 g at 25% of crop evapotranspiration) [23]. According to most authors, fruit weight is strongly affected by environmental conditions, cultivation practices, tree age, and total yield per tree. 

Juice yield, one of the most important parameters from an industrial point of view, was between 37% (cultivar Kamel) and 46% (cultivar Ako), while kernels and peels contributed about 31–40% and 22–23% to the total fruit fw, respectively (Figure 2b). Juice yields between 27 and 55% were reported depending on genotype, pedoclimatic conditions, cultivation practices, and applied processing parameters [24,25,26,27]. Moisture content varied significantly within fruit parts, but not among cultivars (Table 1). On average, moisture was 86.8% in the juice, 31.1% in the kernels, and 76.1% in the peels. Fawole and Opara [28] reported compatible moisture contents in ranges of 64.2–73.91%, 77.9–83.0%, and 74.9–82.0%, respectively, in the rind, mesocarp, and whole arils of seven pomegranate cultivars grown in South Africa. 

Similarly, a study conducted on pomegranate peels from Turkish varieties showed a dry matter content in the range of 24–34% [29]. Unusually low water contents (from 24.12 mg∙g^−1^ to 36.71 mg∙g^−1^) with significant genotype differences were, instead, reported in the kernels of three cultivars (Tunisia soft seed, Taishanhong, and Qingpiruanzi) grown in China by Peng [30]. The pH values and total soluble solids (TSS) of the juice did not vary significantly among cultivars and were, on average, 3.8 and 16.1° Brix, respectively (Table 1). Similarly, peels showed a pH around 3.0–3.5 with no significant cultivar variation, in agreement with the 3.75 value of the peels from Pakistani cultivars [31]. Significant genotypic differences were found, instead, for the contents of total ethanol-soluble carbohydrates and total organic acids in all assayed fruit parts (Table 1). The fresh juices contained 75.2–112.9 mg∙g^−1^ fw total soluble sugars and 15.6–28.8 mg∙g^−1^ fw total organic acids. Meanwhile, the peels and kernels contained between 11.9–40.7 mg∙g^−1^ fw and 46.9–90.0 mg∙g^−1^ fw total ethanol-soluble sugars, respectively, within the ranges reported by other authors [32,33]. In all cultivars, glucose and fructose were the principal monosaccharides identified, with fructose slightly exceeding glucose both in juices and peels (Figure 3a). No sucrose and maltose were detected in any cultivars, in agreement with Fadavi et al. [24]. Generally, pomegranate aril juices have equal amounts of fructose and glucose, with modest variations among cultivars. However, a slight prevalence of the first sugar over the other, and vice versa, was previously reported in the genotypes of different collections [32].

Paradoxically, the sweet cultivar Ako had the lowest total ethanol-soluble carbohydrate content. Accordingly, higher sugar contents in sour cultivars than in sweet ones were reported by Hasnaoui et al. [34] while assaying twelve Tunisian autochthonous genotypes. The same authors described a close correlation between sourness and concentrations of organic acids, especially citric acid. Indeed, total organic acid concentrations in Kamel and Wonderful one Cultivars, which are sweet-sour and sour, respectively, were 73–85% higher than in the sweet ones, with citric acid strongly predominant. Instead, Ako and Emek were characterized by a more balanced organic acid profile (Figure 3b). Differently, malic acid, followed by citric acid, was found as the most abundant organic acid of the juices from the autochthonous wild pomegranates of Montenegro [27]; this discrepancy between wild and cultivated genotypes is likely the result of selective breeding to meet consumer taste preferences. A strong correlation was reported between sourness and citric acid content, which is assumed to be the major factor contributing to pomegranate fruit sourness [34]. Contrary to the juices, the peels showed no detectable amounts of malic acid in all cultivars, while the prevalence of citric acid of sweet-sour and sour cultivars was confirmed. The amount of total ethanol-insoluble carbohydrates (Table 1), mainly comprising starch and dietary fibers (cell wall polysaccharides), was particularly high in kernels (65.1–80.6 mg∙g^−1^ fw), followed by peels (27.9–42.3 mg∙g^−1^ fw) and juice (32.8–35.8 mg∙g^−1^ fw), with no significant differences among cultivars, except for peels. At this regard, it is worth recalling that the exploitation of the polysaccharide fraction potential in the preparation of plant-based products for human health demands deep investigation. Indeed, polysaccharides extracted from pomegranate peels have been proposed to exert immunomodulatory effects and antiproliferative activity on several cancer cell lines [35,36]. 

### 2.2. Functional Quality Attributes 

The total content of phenols, flavonoids, anthocyanins, AsA, and DAsA in the juices, kernels, and peels isolated from the ripe fruits of the four investigated pomegranate cultivars is reported in Table 2. It is worth noticing that, in plants, phenols occur in both soluble and insoluble forms. Soluble phenols are mainly sequestered within vacuoles and are easily extracted from tissues with diluted alcohols (i.e., methanol or ethanol). Insoluble phenols, instead, are covalently bound (primarily esterified) to the polymeric constituents of cell walls, including structural polysaccharides and proteins, and are, thus, resistant to extraction, unless a preliminary hydrolysis step is performed [37].

Though the literature abounds with studies on the phenolic composition of pomegranate arils, juice, and peels, most just focus on soluble phenols, ignoring the insoluble-bound form, often underestimating the total content of phenols in the fruit parts and their health potential. Effectively, in all juice and peel samples, soluble phenols largely exceeded the bound forms, contributing between 97.2 and 99.7% to the total, while they ranged between 66.6 and 76.8% in the kernels. Significant differences were observed among cultivars in both juices and kernels, within a range of 4.6–9.7 and 4.4–6.6 mg gallic acid equivalents (GAE)∙g^−1^ fw, respectively, but not in peels, where the highest concentrations were registered (51.7–61.9 mg GAE∙g^−1^ fw), in agreement with Orak et al. [29]. Similarly, the phenolic content of the peels of pomegranates from three different provinces of Iran (Natanz, Shahreza, and Doorak) was 3.8–11.6- and 5.7–33.3-fold higher than in kernels and juice, respectively [38], and 10-fold higher than that of aril juices from 28 assayed Chinese cultivars [15,39]. 

Of all phenolic compounds, flavanols, ellagitannins and derivatives of ellagic acid had the highest concentrations in pomegranate juices [27], while hydrolysable tannins have been proposed as the predominant class of phenols in peels, with very high amounts of ellagitannins of the gallagyl ester group, such as punicalins and punicalagins [40]. Hence, obtaining juices from whole fruits (peel and arils), as in commercial production, enhances the content and profile of polyphenol compounds and the antioxidant activity of the product [41].

Regarding the juices, our results on total soluble phenolic compounds are generally higher than those reported in the literature, but very close to the contents found for wild Montenegro genotypes (4.4–8.5 mg∙g^−1^ fw) [27,39,42,43]. Lower levels of total soluble phenolic compounds were also reported by Ferrara et al. [22] for juices extracted from Ako (1.77 mg∙g^−1^ fw), Emek (1.50 mg∙g^−1^ fw), and Wonderful One (2.34 mg∙g^−1^ fw) cultivars grown in the same orchard as this study, but in previous years. This suggests that tree age is a factor affecting the level of fruit antioxidants, as was reported for mandarin juice [44]. Reduced levels of total phenolic compounds were also detected by Pande and Akoh [45] and Jing et al. [46] for kernels from Georgia- (0.85–0.91 mg GAE∙g^−1^ fw) and China-grown (1.29–2.17 mg GAE∙g^−1^ fw) pomegranate cultivars, possibly due to different growing conditions, genotypes, and extraction methods. In particular, soluble-conjugated and insoluble-bound phenolic acids were found to significantly contribute to the total phenolic content of kernels, in agreement with Ambigaipalan et al. [47] and Jing et al. [46]. The authors found 1.38, 1.39, and 0.62 mg GAE g^−1^ fw as free, esterified, and insoluble-bound phenolic compounds, respectively, in pomegranate kernels, with 3,4-Dihydroxybenzoic acid predominantly in the soluble-conjugate form, ferulic acid primarily as insoluble-bound form, and both vanillic and syringic acids almost equally distributed between the soluble and insoluble forms. For peels, the levels we found were comparable to the literature data [39]. Kernels, instead, was the fraction most abundant in bound phenols (1.97 to 2.21 mg GAE∙g^−1^ fw), followed by peels (0.19 to 0.23 mg GAE∙g^−1^ fw) and juice (0.07–0.13 mg GAE∙g^−1^ fw), with no statistically significant differences among cultivars. 

Pomegranate fruit is particularly rich in flavonoids in all its parts, hence the interest for an efficient recovery of these bioactive molecules from the industrial by-products [20,48]. Indeed, regardless of genotype, kernels ranked first, also, for total flavonoids (4.2–5.6 mg CE∙g^−1^ fw), followed by peels (4.0–5.1 mg CE∙g^−1^ fw) and juice (0.88–1.44 mg CE∙g^−1^ fw). In agreement, He et al. [49] highlighted the value of a pomegranate seed-residue by-product as a source of several phenolic compounds, including phenolic acid derivatives, flavan-3-ols, flavonoid glycosides, and hydrolysable tannins, and obtained extracts with a total phenolic content up to 2.4 g catechin equivalents (CE)/100 g dry weight (dw) and high antioxidant potential. 

A wide variability of the content of phenols in pomegranate has been reported in the literature, and is strictly related to the cultivar analyzed [50]. According to Li et al. [15], the average total phenolic and flavonoid contents of peels were 249.4 mg tannic acid equivalents (TAE)∙g^−1^ dw and 59.1 mg rutin equivalents (RE)∙g^−1^ dw, respectively. Much lower concentrations, up to 1.22 and 0.13 mg RE∙g^−1^ dw, were, instead, reported by Fellah et al. [51] in the peels of the Nabli and Gabsi Tunisian cultivars, respectively. Our values, expressed on a dw basis after correcting for moisture content (218.1–261.2 mg GAE∙g^−1^ dw and 16.9–21.1 mg CE∙g^−1^ dw, respectively), fall in the upper part of these wide ranges.

The highest anthocyanin concentrations were registered in the juices (217–608 µg cyanidin-3-glucoside equivalents (CGE)∙g^−1^ fw), followed by the peels (104.8–232.9 µg CGE∙g^−1^ fw) and the kernels (61.2–240 µg CGE∙g^−1^ fw). A significant cultivar-dependent variability was found for flavonoids and anthocyanins in all fruit parts. Correspondingly, Elfalleh et al. [52] and Fellah et al. [51] reported significant variation in the total amount of polyphenols, flavonoids, anthocyanins, and hydrolysable tannins among different plant parts (seeds, leaves, flowers, and peels). It has been shown that the bright color of the pomegranate fruit depends mostly on anthocyanin concentration; that, however, can be strongly influenced by several parameters, including pre-harvest treatments (e.g., fertirrigation, methyl jasmonate application), the time of harvesting, and post-harvest storage conditions (temperature, humidity) [52,53,54]. The juice of Wonderful One had levels of anthocyanins considerably higher than those of the other cultivars, presenting, in fact, with an intense deep-red color. Noda et al. [55] reported that three major anthocyanidins found in pomegranate juice were delphinidin, cyanidin, and pelargonidin. Cyanidin 3,5-diglucoside, pelargonidin 3,5-diglucoside, delphinidin 3,5-diglucoside, cyanidin 3-glucoside, pelargonidin 3-glucoside, and delphinidin 3-glucoside were identified in the juices of six old Italian pomegranate varieties, with cyanidin 3-glucoside, followed by pelargonidin 3,5-diglucoside, as the most representative, while anthocyanins were below the limit of detection in the peel extract samples [56]. Only small amounts of anthocyanins (0.37–0.070 mg CGE∙g^−1^ fw) were detected in the kernels of California pomegranates, presumably as contaminants from those remaining from the juiced arils [47]. 

Our results show evident differences in the profiles of the main anthocyanins detected in the juices and peels of the four assayed cultivars (Figure 4). In particular, cyanidin 3- and 3,5-(di)glucosides were present in all samples, although in different quantities and ratios, while delphinidin glucosides were exclusively detected in the juices of the cultivars Kamel and Wonderful One.

With regard to the three fruit parts, the differences observed in the accumulation of phenolics and flavonoids are essentially due to fruit tissue specialization. Peels preferentially accumulate phenolic compounds acting as protective screens against high-energy wavelengths to reduce light-induced oxidative stress and fruit sunburn, as well as those with toxic effects or protein precipitation capacities, as a line of defense against pathogens and predators. The cells of the juicy sarcotesta preferentially synthesize anthocyanins at the cytosolic surface of the endoplasmic reticulum and transport them across the tonoplast through specific carriers (ATP-binding cassette transporter and multidrug and toxic compound extrusion proteins) within large vacuoles. Vacuolar anthocyanins contribute to the brilliant bright-red color of the arils that is fundamental for attracting seed dispersers. Kernels are characterized, instead, by high levels of insoluble-bound phenols likely involved in hardening the thick wall of sclerotesta cells. Phenols (both soluble and insoluble) play, also, vital roles in the protection of the zygotic and reserve tissues from biotic aggression and abiotic stresses favoring seed survival, as well as in seed development, maturation, dormancy, and germination. An increase in the ratio of insoluble-bound to soluble phenolic compounds was observed in lentil seeds during germination, suggesting that phenol secretion to the cell wall, followed by insolubilization, may exert a pivotal action in regulating the process [37,57].

AsA and DAsA contents of most fruit parts were significantly affected by genotype. Peels revealed the highest concentration of both AsA (2.6–4.4 mg∙g^−1^ fw) and DAsA (0.3–2.4 mg∙g^−1^ fw), followed by juices (0.42–0.95 and 0.06–0.70 mg∙g^−1^ fw) and kernels (0.22–0.37 and 0.03–0.16 mg∙g^−1^ fw) (Table 2). In agreement with our results, Opara et. al. [58] reported that vitamin C content in the pomegranate peel was significantly higher than in the aril, with differences ranging from 24.4% to 97.0%, depending on variety. Compared to our results, much lower levels of ascorbic acid were found in pomegranate juices from India (0.198 mg g^−1^ fw), Iran (0.09–0.40 mg g^−1^ fw), Poland (0.091–0.268 mg g^−1^ fw), and South Africa (0.168 mg g^−1^ fw) [23,40,59,60]. This discrepancy is possibly due to different agricultural practices and/or to the peculiar pedoclimatic conditions of southern Italy. In particular, the mild weather, high sun irradiation, and the medium-textured lime-rich soils of the Salento peninsula make for the perfect terroir for the development and ripening of some fruits with improved levels of primary and secondary metabolites, including grapes, tomatoes, and olives [61,62,63]. 

Interestingly, pomegranate peels had, on average, 12-fold higher AsA concentrations than the peels of oranges, limes, and mandarins, fruits notoriously rich in vitamin C, while the AsA contents of the juices were similar or even higher than in the pulp of citrus fruits, as well as of pineapple, kiwi, and other pomegranate cultivars [64,65]. Nevertheless, it is important to reiterate that vitamin C is very unstable, being sensitive to light, heat, and air, rapidly deteriorating in the food during transport, processing, denting, and cutting [66].

Therefore, the obtained results suggest the need to raise the consumer’s awareness about the importance of pomegranate as a source of vitamin C that widely exceeds other consumed fresh products. 

### 2.3. Kernel Lipid Content and Fatty Acid Profiles 

The kernels’ total lipids showed low variability among the assayed Israeli genotypes, with Ako giving the highest yield (255 mg∙g^−1^ fw), and all other cultivars giving an average of 174 mg∙g^−1^ fw (Table 3). 

These findings are in agreement with previous reports on Israeli genotypes grown in the same orchard and generally fall into the mid-upper range of values obtained in comparative studies on large selections of pomegranate cultivars from different origins [21,29,67,68], thus confirming a possible convenience of extracting oil from their kernels.

The fatty acid composition of the kernels showed significant differences among cultivars. Polyunsaturated fatty acids (PUFA) were the most abundant (86.71–89.36% of total identified fatty acids), followed by saturated (SFA, 3.90–4.57%) and monounsaturated acids (MUFA, 6.74–8.72%), in good agreement with the literature [45,69]. In all cultivars, punicic acid (C18:3 n-5), a conjugated isomer of α-linolenic acid, was found in high amounts (70.23–77.65%), in agreement with several reports on cultivars grown in some of the top producer countries (Brazil, China, India, Israel, Tunisia, Turkey, and the USA), in which its level varied between 37 and 81% [29,70,71]. This is of particular interest, as punicic acid exerts important bioactivities, including anticancer, hypolipidemic, antidiabetic, and antiobesity [72]. Punicic acid is synthesized in situ from linoleic acid, found in lower amounts (between 4.64 and 5.27%) in the kernel oil. To a lesser extent, oleic (3.90–4.57%), palmitic (3.26–4.57%), and stearic (3.15–4.15%) acids, as well as other C18:3 *n*-3 isomers (7.07–12.12%), were also found in all cultivars (Table 3).

### 2.4. Antioxidant Activities 

The HAA, LAA, and TAA of juices, kernels, and peels isolated from each cultivar are shown in Table 4.

HAA largely exceeded LAA in all fruit parts, with both varying significantly among cultivars. Generally, peels had the highest HAA and LAA values (294.0–368.8 and 43.9–74.8 mM Trolox equivalents (TE)∙g^−1^ fw, respectively), whereas lower values were obtained for juices (51.2–77.3 and 0.2–1.4 mM TE∙g^−1^ fw) and kernels (23.3–45.2 and 1.7–4.1 mM TE∙g^−1^ fw). Lower ABTS inhibition rates, within 1.61–2.53 and 7.8–15.81 μmol TE∙g^−1^, were reported by Peng [29] and Jing et al. [45] for hydroalcoholic extracts from the kernels or defatted kernel flours of Chinese and Georgia-grown pomegranate varieties. The HAA of peel homogenates from a broad, bio-diverse pomegranate collection from different countries was up to 40-fold higher than that measured in the aril juice, and directly correlated to the levels of total phenolic compounds in all fruit parts [73]. In agreement, a strong linear correlation between TEAC values and total soluble phenolics was obtained in the present study (Table 5). Correlation was also found for antioxidant activities versus AsA and DAsA, indicating their synergistic contributions to the antioxidant activity of pomegranate fruits.

### 2.5. Element Composition

Table 6 summarizes the elemental concentrations of the different fruit parts isolated from ripe fruits of the four pomegranate cultivars under investigation. The amount of most elements was significantly different among either cultivars or fruit parts. Overall, potassium (K) ranked first for concentration, followed by magnesium (Mg) and sodium (Na). The relative order of the other assayed elements was, instead, variable within samples. Boron (B), iron (Fe), and copper (Cu) were the main microelements in the juices, kernels, and peels, respectively. Fe, Cu, Na, Mg, manganese (Mn), vanadium (V), and zinc (Zn) concentrations were significantly lower in juices than in peels and, largely, in kernels, where the highest concentrations were registered on a fw basis regardless of cultivar. Similarly, aluminum (Al) content was low in the juices of all cultivars, but much higher in the other fruit parts, with kernels or peels showing the highest concentrations depending on cultivar. Silver (Ag) and molybdenum (Mo) were, instead, only detected in the peels of Ako and Kamel, respectively, and in the kernels of all cultivars, suggesting a selective sequestration of these elements in the different fruit tissues. 

Ag, bismuth (Bi), cadmium (Cd), cobalt (Co), lithium (Li), and thallium (Tl) were, instead, below the detection limit (0.001 µg∙g^−1^ fw) in all samples. Most of our findings are consistent with the results of other studies on the mineral composition of pomegranates grown in different countries. Indeed, many authors reported that the main elements found in the juice of pomegranates were, roughly in decreasing order, nitrogen (N), K, Na, phosphorous (P), Mg, calcium (Ca), Fe, Zn, Cu and Mn, within concentration ranges that were largely affected by genotype and ripening stage, as well as by pedological, climatic, and agronomical factors [27,74,75,76,77,78]. Recently, Loukhmas et al. [79] found iodine (I) as the most abundant macroelement in pomegranate juice obtained from ten Moroccan pomegranate cultivars, followed by phosphorus (P) and sulfur (S).

According to Mirdehghan and Rahemi [80], the relative order of concentration of macroelements both in the arils and peels was K > N > Ca > P > Mg > Na, while the relative order of concentration of microelements was B > Fe > Zn > Cu > Mn in the arils and B > Fe > Zn = Mn > Cu in the peels of freshly harvested pomegranate fruits. A similar order (Na > > K >>Fe > Mn-Zn) was reported by Ullah et al. in the peels of an unspecified pomegranate cultivar grown in Pakistan [30]. Fawole and Opara [27] found considerable differences in trace element concentrations in the fruit parts (rind, mesocarp, and entire arils) and among cultivars in seven pomegranate genotypes (Arakta, Bhagwa, Ganesh, Herskawitz, Mollar de Elche, Ruby, and Wonderful) grown in South Africa. The authors reported relatively high amounts of Mn, Fe, Cu, Zn, B, Nickel (Ni), Selenium (Se), Al, and Strontium (Sr) in most cultivars, while the amounts of Ni, Co, Pb, Cd, As, Li, Ti, and V in the fruit parts were low or below the detection level. They also found that Na was particularly abundant in the cultivar Wonderful; accordingly, we found that this element was much more concentrated in the juices and peels of the related genotype Wonderful One than in all other cultivars. Further, Okatan et al. [78] reported a statistically significant positive correlation between N and Cu contents, as well as between phosphorus, AsA, TAA, and total anthocyanins. 

### 2.6. Principal Component Analysis

To highlight biochemical differences among fruit parts and genotypes and to highlight any clustering of the observations, a multivariate analysis (PCA) was carried out (Figure 5). 

Two relevant principal components (PCs) that explain up to 81.57% of the variation of the collected data were extracted. The first axis (PC1) explains 52.09% of the total variance, while the second (PC2) explains the remaining 29.48%. The contribution of each quality parameter (variables) is reported in Table 7. 

The PC1 vs. PC2 plot shows a clear clustering of fruit parts. Irrespectively of cultivars, all juices grouped on the lower-left side of the chart show high moisture and high concentrations of soluble carbohydrates and anthocyanins. Anthocyanin levels in the pomegranate fruit are developmentally regulated. Yuan, et al. [81] identified 26 candidate genes involved in the biosynthesis of anthocyanins, which showed tissue-specific expression in arils and peels. Indeed, high anthocyanin levels in both fruit parts are a key factor for varietal selection because they largely determine the economic value of the fruit. Kernels were, instead, distributed in the mid-right side of the chart, being characterized by high levels of metallic minerals, ethanol-insoluble carbohydrates, and insoluble-bound phenolics. The close correlation between these variables confirms that insoluble phenols are mainly bound to structural polysaccharides, which, in turn, play a role in accumulating/sequestering metal ions contributing to cell wall architecture or those potentially toxic to the protoplast. Boron, for example, is an important structural element of plant cell walls as it is involved in rhamnogalacturonan-II crosslinking. 

Peels grouped in the upper left pane show large amounts of soluble phenols, AsA, and DAsA, as well as high antioxidant activities. This possibly reflects the protective role that peel phenolic compounds exert against solar UV radiations and/or pathogen infections. 

## 3. Materials and Methods

### 3.1. Chemicals

ABTS (2,2′-azinobis-[3-ethylbenzthiazoline-6-sulfonic acid] diammonium salt), Trolox (6-Hydroxy-2,5,7,8-tetramethylchromane-2-carboxylic acid), catechin, Folin and Ciocalteu’s phenol reagent, sodium carbonate (Na_2_CO_3_), gallic acid, vanillin, catechin, sodium nitrite (NaNO_2_), aluminum chloride hexahydrate (AlCl_3_ × 6 H_2_O), DTT (dithiothreitol), NEM (N-ethylmaleimide), 2,2′-dipyridyl, iron chloride hexahydrate (FeCl_3_ x 6 H_2_O), ascorbic acid, as well as all HPLC-grade solvents were purchased from Sigma–Aldrich (Milan, Italy). High purity standards for the qualitative-quantitative determination of fatty acids (palmitic, stearic, oleic, linoleic, punicic), organic acid (citric, malic, succinic), sugars (glucose and fructose) were purchased from Sigma–Aldrich (Milan, Italy). Anthocyanin (cyaniding 3-glucoside, cyaniding 3,5-diglucoside, delphinidin 3-glucoside, delphinidin 3,5-diglucoside) standards were purchased from Extrasynthase (GenayCedex, France). 

### 3.2. Plant Material and Growth Conditions

The analyses were carried out on pomegranate fruits of four Israeli cultivars (Ako, Emek, Kamel, and Wonderful One) randomly harvested from different trees, cultivated at the Cairo & Doutcher Farm (40.298757,18.022788) in Copertino (Lecce, Italy), during the 2016 production season (from mid-September to the end of October). 

The average annual climate parameters of the experimental region are: temperature, 17 °C (with a maximum of 30.2 °C in August and a minimum of 9.9 °C in January); precipitation, 606–640 mm; relative humidity, 74–61%; average UV index, 7–4. The soil is a red earth that is dark in color, moderately to very deep, with little structure, and a clay-loam texture classified as Alfisol according to the US Soil Taxonomy.

The plants, aged 5 years, were arranged in the field at a distance of 6 m between the rows and 2.5 m in the row and supported by galvanized iron Y-shaped poles holding two lines of steel wires on each arm. Mulching was made on the row with white plastic film (thickness 100 mm, width 1.5 m), tucked up by 25–30 cm on both sides and with about 1 m of free span to reduce weeds and water consumption and to increase the refraction of sunlight towards the canopy. A complete fertigation system, with a dripline of 16 mm per row, with drips at 50 cm and a capacity of 2.2 L‧h^−1^, was used with an average total water consumption for one year of 3500 m^3^‧ha^−1^. Fertigation was applied with nitrogen, phosphorous, and potassium (250, 50–60, and 250 Units‧ha^−1^ per year, respectively, split for the entire production cycle) and chelated iron (3 applications per year of 50 kg‧ha^−1^ each). Brown pruning was carried out in the dormant season (January) and green pruning was carried out after the fruit was set (July) for sucker removal and thinning. Aphid control was carried out using diluted liquid soap or white oil, when required. 

Healthy, undamaged (with no cracking and intact calix) samples were harvested at the commercial ripening stage based on the visual assessment of the skin color and of the overall fruit appearance. The pomegranate collection included two sweet (Ako and Emek), one sweet-sour (Kamel), and one sour (Wonderful One) genotypes (Figure 2). 

Freshly harvested fruits (at least 3 kg for each cultivar) were externally washed with tap water, dried, cut into halves, and pressed by a professional electric lever squeezer (Fimar s.r.l., Rimini, Italy) to recover the juice. Kernels were manually separated from the marc (the residue of the fruit after squeezing), deprived of the debris of sarcotesta by rubbing them on filter paper, dried at room temperature under a fume board, and ground in a laboratory ultra-centrifugal mill (ZM200, Retsch, Haan, Germany) through a 35 mesh (500 µm) sieve. The remaining peels (rinds plus albedo) were cut into small pieces and homogenized under liquid nitrogen in a high-speed blender (Waring Laboratory Science, Torrington, CT, USA). Peel and kernel samples were vacuum-packaged in food-grade oxygen-impermeable plastic bags, while pomegranate juice was stored in hermetic PET bottles. All fruit parts were stored at −80 °C until analyses. 

### 3.3. Determination of Moisture, TSS, and pH

Moisture was determined gravimetrically after drying 1.0 g aliquots of each fruit part at 105 °C in a Büchi TO-50 infrared dryer (BÜCHI Labortechnik AG, Flawil, Switzerland). The juice TSS, expressed as °Brix, and the pH were determined at room temperature using a digital refractometer (DBR95 Giorgio Bormac S.r.l., Carpi (Modena), Italy) and a pH meter (Mettler Toledo, Columbus, OH, USA), respectively. Ash content was determined for 5.0 g aliquots of each fruit part using a muffle furnace at 525 °C for 6 h, according to the AOAC (2005) method. The ash content was recorded as g per 100 g fw (g∙100 g^−1^ fw).

### 3.4. Determination of Carbohydrate and Organic Acid Contents

Total ethanol-soluble and ethanol-insoluble carbohydrates were determined with the phenol-sulfuric acid method using a Beckman DU 650 spectrophotometer (Beckman Coulter Inc., Brea, CA, USA) according to Nielsen [82] on the surnatants and precipitates, respectively, obtained by mixing appropriate aliquots of each fruit part (5 mL juice, 1.0 g kernels or peels) with 70% ethanol and centrifuging at 6000 g (10 min). Quali-quantitative profiles of soluble sugars and organic acids were simultaneously carried out according to Tufariello et al. [83] with a 1100 Series HPLC system (Agilent Technologies Inc., Santa Clara, CA, USA) equipped with an Aminex HPX-87H column (300 × 7.8 mm) (Biorad, Hercules, CA, USA). 

### 3.5. Extraction and Analysis of Soluble and Insoluble-Bound Phenolic Compounds

Extraction of soluble and insoluble-bound phenolic compounds was carried out according to the method reported by Durante et al. [17]. The levels of phenolic compounds of each extract were determined by the Folin–Ciocalteau method [84]. Briefly, 50 µL of extract was mixed with 50 µL of Folin–Ciocalteau phenol reagent and 450 µL distilled water. After 5 min, 500 µL of 7% Na_2_CO_3_ (*w*/*v*) and 200 µL of distilled water were added. The mixture was left at room temperature in the dark for 90 min. The absorbance was read at 750 nm with a Beckman DU650 spectrophotometer (Beckman Coulter Ltd., High Wycombe, UK). Total phenolic contents were calculated based on the calibration curve of gallic acid (GA) (0–12 µg GA∙100 µL^−1^). The results were expressed as mg GA equivalents (GAE)∙g^−1^ fw.

### 3.6. Extraction and Determination of Flavonoids

To 0.3 g of each fruit part, 1.5 mL of 100% methanol (*v*/*v*) was added. The mixture was then shaken at 4 °C for 16 h and centrifuged at 8800× *g* for 10 min. All supernatants were recovered and used for determining total flavonoid and condensed tannin contents.

The total flavonoid content was determined as described by Zhishen et al. [85]. Briefly, 50 µL of extract was diluted with 450 µL distilled water and 30 µL of 5% NaNO_2_ (*w*/*v*). After 5 min, 60 µL of 10% AlCl_3_ hexahydrate (*w*/*v*) was added, followed by, after a further 6 min, 200 µL of 1 M NaOH and 210 µL of distilled water. The absorbance was read at 510 nm with a Beckman DU650 spectrophotometer. The total flavonoid content was calculated on the basis of the calibration curve of catechin (0 to 400 µg catechin mL^−1^) and expressed as mg of catechin equivalents (CE)∙g^−1^ fw.

### 3.7. Extraction and Quantification of Anthocyanins

Extraction of total anthocyanins was carried out according to Zhao et al. [86]. Briefly, 0.5 g of each fruit part was mixed with 15 mL of methanol containing 0.1% HCl. The samples were stirred at room temperature in the dark for 30 min. Subsequently, the samples were centrifuged at 3900× *g* for 30 min at 4 °C in a Beckman Allegra^TM^ X-22 centrifuge. All supernatants were recovered and dried by evaporation at a constant temperature of 35 °C. The dry residue was suspended in 1 mL of distilled water. 

Total anthocyanin content was estimated by the pH differential method using two buffer systems: potassium chloride buffer pH 1.0 (25 mM) (buffer 1) and sodium acetate buffer pH 4.5 (0.4 M) (buffer 2) [87]. Two aliquots (0.1 mL) of the extract from each fruit part were alternatively mixed with 0.9 mL of buffer 1 or buffer 2. The samples were incubated at room temperature (in the dark, for 15 min), and read against water at 510 and 700 nm. The absorbance was calculated as follows: A = (A_510 nm_ − A_700 nm_)pH_1.0_ − (A_510 nm_ − A_700 nm_)pH_4.5_, with a molar extinction of 26,900. The results were expressed as µg of cyanidin-3-glucoside (CGE)∙g^−1^ fw.

Anthocyanin extracts were analyzed quali-quantitatively, as described by Laddomada et al. [88], using an Agilent 1100 Series HPLC system equipped with a Luna 5 μm C18(2) 100 Å column (250 × 4.6 mm) (Phenomenex, Torrance, CA, USA). The wavelength used for quantification of anthocyanin compounds was 520 nm.

### 3.8. Extraction and Determination of Ascorbic Acid (AsA) and Dehydroascorbic Acid (DHA) Contents

Extraction and determination of AsA and DHA were carried out according to Kampfenkel et al. [89] on 0.2 g of each fruit part. The absorbance was read at 525 nm in a Beckman DU650 spectrophotometer. The linear curve was from 0 to 600 µM AsA. The results were expressed as mg∙g^−1^ fw.

### 3.9. Determination of Hydrophilic (HAA) and Lipophilic (LAA) Antioxidant Activities

HAA and LAA were determined by the TEAC (Trolox Equivalent Antioxidant Capacity) assay [90], slightly modified by Durante et al. [91]. Antioxidant compounds were sequentially extracted from 0.4 g of each fruit part with 100% methanol (hydrosoluble antioxidants) and 100% acetone (liposoluble antioxidants) under constant shaking (300 rpm) at room temperature in the dark for 1 h. Samples were centrifuged at 2500× *g* for 10 min. Supernatants were recovered and used for measurements. The scavenging capacity against ABTS^• +^ (absorbance decrease) was measured at 734 nm in a Beckman DU650 spectrophotometer. The linear calibration curves were from 0 to 15 µM Trolox for both HAA and LAA. TAA was calculated as the sum of HAA and LAA. Results were expressed as µmol of TE∙g^−1^ fw.

### 3.10. Lipids Content and Composition of Fatty Acids

Kernel oil was extracted by petroleum ether using a Soxhlet for 6 h, according to the 30–25.01 AACC method [92], in triplicate for each cultivar, and expressed as mg∙g^−1^ kernel fw. 

For fatty acid analysis, lipids were extracted from 100 mg of kernels with 5 mL of *n*-hexane under mechanical stirring (300 rpm) at 4 °C for 16 h. Samples were centrifuged at 4500 *g* for 5 min and the organic phase was recovered and evaporated to dryness under a stream of nitrogen. Fatty acid derivatization and analyses were carried out according to Durante et al. [91] using an Agilent 5977E GC/MS system equipped with an Agilent DB-WAX column (60 m, 0.25 mm i.d., 0.25 mm film thickness).

### 3.11. Element Assay

Element concentrations were measured using an inductively coupled plasma atomic emission spectrometer (ICP-AES; Thermo Fisher Scientific iCap 6000 Series) as reported in Bruno et al. [93]. Results were expressed as mg∙g^−1^ fw.

### 3.12. Statistical Analysis and Multivariate Data Analysis

Results are presented as the mean value ± standard deviation of three independent biological replicate experiments (*n* = 3; for each cultivar, three different batches of fruits (at least 3 kg from the same harvesting lot) were separately processed and analyzed). Statistical analysis was based on a one-way ANOVA test. The Holm–Sidak post hoc method was applied to establish significant differences between means (*p* < 0.05). Correlations were calculated using Pearson’s correlation coefficient (*r*). Statistical comparisons were performed using SigmaStat software, version 11.0 (Systat Software Inc., Chicago, IL, USA). 

PCA was performed on the complete data matrix of each pomegranate cultivar using the XLSTAT software (Addinsoft, Paris, France). The analysis was carried out by plotting the mean values of each evaluated parameter (variables) within juices, kernels, and peels isolated from the fruits of each cultivar. 

## 4. Conclusions

In summary, a significant genotype-dependent variability was observed for many of the parameters investigated in all fruit parts. However, no clear relationships were evidenced between the concentrations of each analyzed class of compounds within the different fruit parts. Thus, for example, the juices of Ako, Emek, and Wonderful One had the highest content of total soluble phenolic compounds, while in kernels, the cultivar Kamel ranked first for the same parameter, and no significant variations were detected among genotypes in peels. Interestingly, the total antioxidant activity of all fruit parts linearly correlated with the amounts of total soluble phenolic compounds, AsA, and DAsA, indicating their synergistic contribution to the antiradical properties of the juices, as well as of the kernel and peel extracts. 

The obtained values for most antioxidants within each fruit part were, generally, higher than the literature data. Such results can be possibly explained by the suitable and prevailing pedoclimatic factors of the Salento peninsula, to which the studied genotypes seem perfectly adapted. This discrepancy suggests that the interaction between genetic background, geographical growing location, and agronomic factors strongly contributes to shaping the functional quality of pomegranate fruit and highlights a hidden genotypic potential in accumulating bioactive molecules that should be revealed.

The results of the present study also confirm that pomegranate by-products are a rich source of bioactive molecules and elements with potential human health benefits. Despite showing significant quali-quantitative differences, kernels of all assayed cultivars are a non-conventional resource to produce edible oil that is particularly rich in essential conjugated linolenic acids and other biologically active compounds. Peels, instead, are an excellent reservoir of different classes of phenolic compounds exerting strong antioxidant activity. However, given the wide variability found, further research on the different fruit parts from each genotype is essential to best exploit their agro-industrial potential and increase sustainability by reducing wastes from the food-production chains.

This is the first systematic research on the quality and functional attributes of different fruit parts from pomegranate cultivars grown simultaneously in the same orchard and subjected to identical agronomic and environmental conditions.

Overall, the study provides greater knowledge of several characteristics contributing to the health benefits and marketing of pomegranate fruits that can be used to assist breeders and growers to respond to consumer and industrial preferences, as well as encourage the development of biorefinery strategies for the utilization of pomegranate by-products as nutraceuticals or value-added ingredients for custom-tailored supplemented foods. 

## Figures and Tables

**Figure 1 plants-10-02521-f001:**
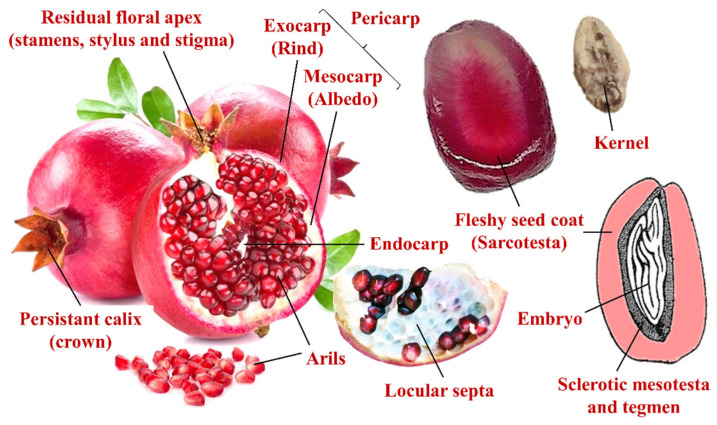
Anatomy of a typical pomegranate fruit and seed (aril).

**Figure 2 plants-10-02521-f002:**
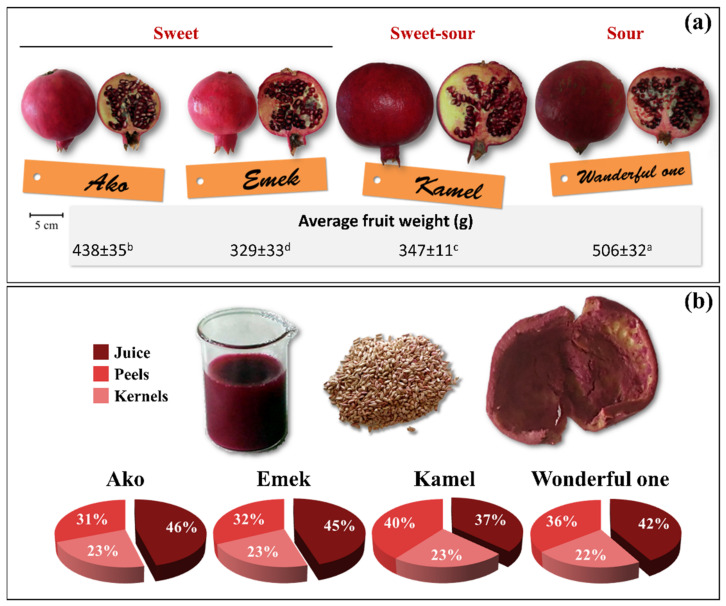
(**a**) External appearance, longitudinal sections, and average weight (±standard deviation) of ripe fruits of different pomegranate cultivars grown in an open field in the south of Italy. Fruit weight data were submitted to one-way analysis of variance (ANOVA) and differences among cultivars were detected using multiple comparison procedures (Holm–Sidak post hoc test, *n* = 20, *p* < 0.05) and indicated with different superscript letters. (**b**) Macroscopic appearance of the juice, kernel, and peel fruit parts isolated from a representative fruit (cultivar: Wonderful One) and the relative average percent weights for each cultivar under analysis.

**Figure 3 plants-10-02521-f003:**
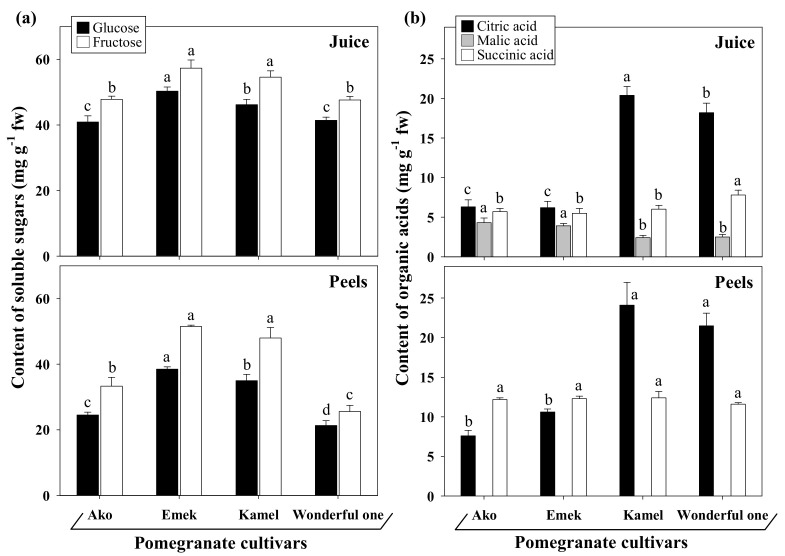
Profiles of the main soluble sugars (**a**) and organic acids (**b**) detected in the juices and peels isolated from the ripe fruits of different pomegranate cultivars grown in an open field in the south of Italy. Values represent the mean ± standard deviation of three independent sampling replicates (*n* = 3). Data were submitted to one-way analysis of variance (ANOVA) and differences among cultivars, within each category, were detected using multiple comparison procedures (Holm–Sidak post hoc test, *p* < 0.05) and indicated with different letters.

**Figure 4 plants-10-02521-f004:**
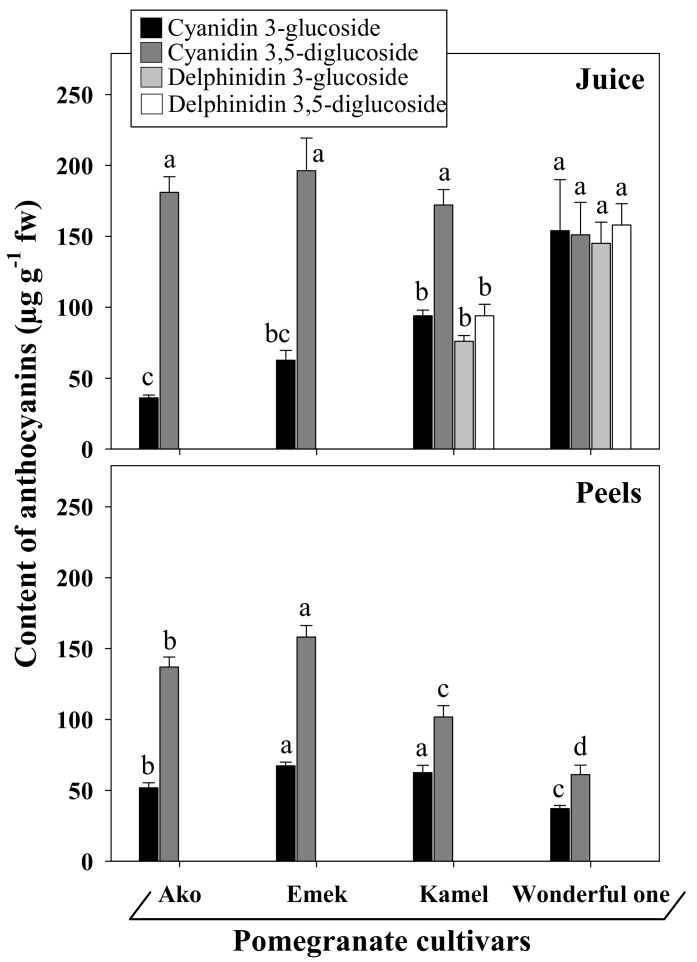
Profile of the main anthocyanins detected in the juices and peels isolated from the ripe fruits of different pomegranate cultivars grown in an open field in the south of Italy. Values represent the mean ± standard deviation of three independent sampling replicates (*n* = 3). Data were submitted to one-way analysis of variance (ANOVA) and differences among cultivars, within each category, were detected using multiple comparison procedures (Holm–Sidak post hoc test, *p* < 0.05) and indicated with different letters.

**Figure 5 plants-10-02521-f005:**
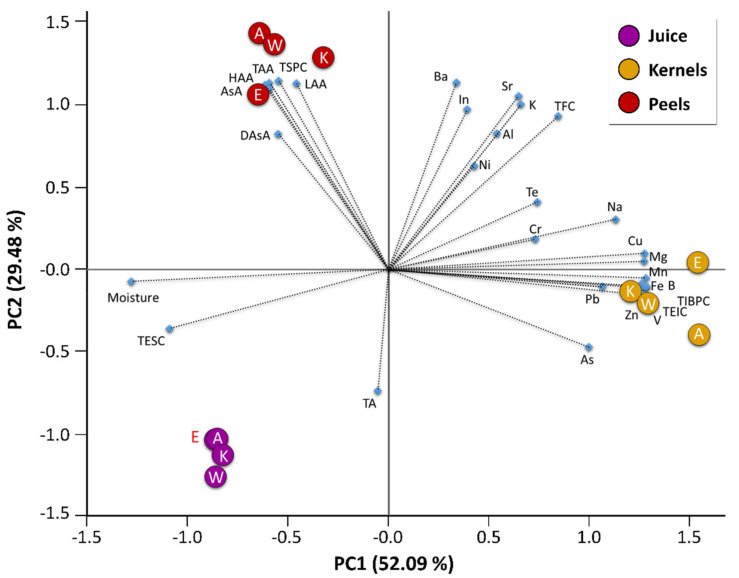
Principal component analysis (PCA) biplot of PC1 vs. PC2 of fruit quality attributes (moisture, total ethanol-soluble carbohydrates (TESC)), total ethanol-insoluble carbohydrates (TEIC)), functional quality attributes (total soluble phenolic compounds (TSPC), total insoluble-bound phenolic compounds (TIBPC), total flavonoid compounds (TFC), total anthocyanins (TA), ascorbic acid (AsA), dehydroascorbic acid [DAsA), antioxidant activities (hydrophilic antioxidant activity (HAA), lipophilic antioxidant activity (LAA), total antioxidant activity (TAA)), and element composition of different fruit parts (juice, kernels and peels) isolated from the ripe fruits of different pomegranate cultivars grown in an open field in the south of Italy. The variance (%) explained by each PCA axis is given in brackets. The length of the vectors is correlated to their significance within each population. Between vectors and between a vector and an axis, there is a positive correlation if the angle is <90°, whereas the correlation is negative if the angle reaches 180°. There is no linear dependence if the angle is 90°. A, Ako; E, Emek; K, Kamel; W, Wonderful One.

**Table 1 plants-10-02521-t001:** Chemical characterization (moisture, pH, TSS, carbohydrates, and total organic acids) of juices, kernels, and peels isolated from the ripe fruits of different pomegranate cultivars grown in an open field in the south of Italy. Values represent the mean ± standard deviation of three independent sampling replicates (*n* = 3). Data were submitted to one-way analysis of variance (ANOVA) and differences among groups were detected using multiple comparison procedures (Holm–Sidak post hoc test, *p* < 0.05). For each trait, values within the column followed by different lowercase superscript letters denote significant differences among cultivars, while values within the rows followed by different uppercase superscript letters denote significant differences among parts.

	Fruit Parts		
	Juice	Kernels	Peels
**Moisture (%)**			
Ako	86.5 ± 0.5 ^aA^	31.0 ± 1.2 ^aC^	75.8 ± 0.4 ^aB^
Emek	87.0 ± 0.5 ^aA^	31.1 ± 1.0 ^aC^	76.3 ± 0.3 ^aB^
Kamel	86.3 ± 0.4 ^aA^	30.8 ± 1.0 ^aC^	76.3 ± 0.2 ^aB^
Wonderful One	87.3 ± 0.4 ^aA^	31.3 ± 1.1 ^aC^	76.1 ± 1.1 ^aB^
**pH**			
Ako	4.1 ± 0.2 ^aA^	nd	3.5 ± 0.5 ^aA^
Emek	4.1 ± 0.3 ^aA^	nd	3.5 ± 0.5 ^aA^
Kamel	3.6 ± 0.2 ^aA^	nd	3.0 ± 0.5 ^aA^
Wonderful One	3.5 ± 0.2 ^aA^	nd	3.0 ± 0.5 ^aA^
**TSS (°Brix)**			
Ako	15.9 ± 0.3 ^aA^	nd	5.5 ± 0.1 ^abB^
Emek	16.0 ± 0.3 ^aA^	nd	5.0 ± 0.3 ^bB^
Kamel	16.3 ± 0.3 ^aA^	nd	5.9 ± 0.2 ^aB^
Wonderful One	16.3 ± 0.4 ^aA^	nd	5.1 ± 0.1 ^bB^
**Total ethanol-soluble carbohydrates (mg∙g^−1^ fw)**			
Ako	75.2 ± 11.3 ^bA^	40.7 ± 2.3 ^aC^	57.8 ± 3.6 ^bB^
Emek	105.2 ± 8.2 ^aA^	26.3 ± 0.2 ^bC^	90.0 ± 1.1 ^aB^
Kamel	100.4 ± 2.1 ^aA^	18.9 ± 0.5 ^cC^	83.0 ± 5.1 ^aB^
Wonderful One	112.9 ± 8.9 ^aA^	11.9 ± 2.6 ^dC^	46.9 ± 3.4 ^cB^
**Total ethanol-insoluble carbohydrates (mg∙g^−1^ fw)**			
Ako	33.9 ± 4.2 ^aB^	80.6 ± 24.1 ^aA^	34.1 ± 0.7 ^bB^
Emek	35.8 ± 4.1 ^aB^	70.3 ± 26.4 ^aA^	42.3 ± 1.1 ^aB^
Kamel	33.2 ± 4.5 ^aC^	70.5 ± 12.4 ^aA^	40.7 ± 0.8 ^aB^
Wonderful One	32.8 ± 3.2 ^aB^	65.1 ± 8.3 ^aA^	27.9 ± 0.9 ^cB^
**Total organic acids (mg∙g^−1^ fw)**			
Ako	16.3 ± 1.9 ^bB^	nd	19.8 ± 0.9 ^bA^
Emek	15.6 ± 1.7 ^bB^	nd	22.9 ± 0.7 ^bA^
Kamel	28.8 ± 1.9 ^aB^	nd	36.5 ± 3.7 ^aA^
Wonderful One	28.5 ± 2.1 ^aB^	nd	33.1 ± 1.8 ^aA^

nd, not determined; TSS, total soluble solids.

**Table 2 plants-10-02521-t002:** Antioxidant composition (soluble and insoluble-bound phenolics, total flavonoids, total anthocyanins, AsA, and DAsA) of the juices, kernels, and peels isolated from the ripe fruits of different pomegranate cultivars grown in an open field in the south of Italy. Values represent the mean ± standard deviation of three independent sampling replicates (*n* = 3). Data were submitted to one-way analysis of variance (ANOVA) and differences among groups were detected using multiple comparison procedures (Holm–Sidak post hoc test, *p* < 0.05). For each trait, values within the column followed by different lowercase superscript letters denote significant differences among cultivars, while values within the rows followed by different uppercase superscript letters denote significant differences among fruit parts.

	Fruit Parts		
	Juice	Kernels	Peels
**Soluble phenolics (mg GAE∙g^−1^ fw)**			
Ako	9.6 ± 0.3 ^aB^	4.4 ± 0.2 ^bC^	61.4 ± 5.3 ^aA^
Emek	9.7 ± 0.6 ^aB^	5.1 ± 0.2 ^abC^	51.7 ± 0.1 ^aA^
Kamel	4.6 ± 0.7 ^bC^	6.6 ± 0.6 ^aB^	61.9 ± 0.2 ^aA^
Wonderful One	8.7 ± 0.2 ^aB^	5.0 ± 0.3 ^abC^	52.3 ± 3.7 ^aA^
**Insoluble-bound phenolics (mg GAE∙g^−1^ fw)**			
Ako	0.07 ± 0.01 ^aC^	2.21 ± 0.1 ^aA^	0.23 ± 0.04 ^aB^
Emek	0.09 ± 0.01 ^aC^	1.97 ± 0.1 ^aA^	0.21 ± 0.03 ^aB^
Kamel	0.13 ± 0.01 ^aC^	1.99 ± 0.1 ^aA^	0.20 ± 0.00 ^aB^
Wonderful One	0.12 ± 0.06 ^aB^	2.08 ± 0.1 ^aA^	0.19 ± 0.04 ^aB^
**Total flavonoids (mg CE∙g^−1^ fw)**			
Ako	0.88 ± 0.04 ^cC^	4.2 ± 0.5 ^bB^	5.1 ± 0.1 ^aA^
Emek	1.44 ± 0.03 ^aC^	5.4 ± 0.4 ^aA^	4.0 ± 0.0 ^cB^
Kamel	0.94 ± 0.01 ^cC^	5.6 ± 0.1 ^aA^	4.4 ± 0.2 ^bcB^
Wonderful One	1.27 ± 0.02 ^bB^	4.8 ± 0.3 ^abA^	4.7 ± 0.1 ^abA^
**Total anthocyanins (µg CGE∙g^−1^ fw)**			
Ako	217 ± 13 ^bA^	61.2 ± 11.8 ^bC^	173.2 ± 19.3 ^abB^
Emek	259 ± 30 ^bA^	132.0 ± 62.0 ^bA^	232.9 ± 14.4 ^aA^
Kamel	436 ± 27 ^abA^	101.8 ± 38.3 ^bC^	178.5 ± 15.8 ^abB^
Wonderful One	608 ± 88 ^aA^	240.8 ± 20.2 ^aB^	104.8 ± 12.7 ^bC^
**AsA (mg∙g^−1^ fw)**			
Ako	0.95 ± 0.03 ^aB^	0.37 ± 0.03 ^aC^	4.4 ± 0.0 ^aA^
Emek	0.58 ± 0.02 ^cB^	0.37 ± 0.05 ^aC^	2.6 ± 0.1 ^bA^
Kamel	0.42 ± 0.02 ^dB^	0.30 ± 0.05 ^aC^	3.0 ± 0.3 ^bA^
Wonderful One	0.71 ± 0.03 ^bB^	0.22 ± 0.02 ^aC^	4.0 ± 0.2 ^aA^
**DAsA (mg∙g^−1^ fw)**			
Ako	0.23 ± 0.02 ^bB^	0.16 ± 0.02 ^aC^	2.4 ± 0.1 ^aA^
Emek	0.70 ± 0.01 ^aB^	0.13 ± 0.01 ^aC^	1.8 ± 0.1 ^bA^
Kamel	0.20 ± 0.03 ^bC^	0.03 ± 0.01 ^bB^	0.3 ± 0.0 ^dA^
Wonderful One	0.06 ± 0.01 ^cB^	0.07 ± 0.00 ^bB^	0.9 ± 0.0 ^cA^

GAE, gallic acid equivalents; CE, catechin equivalents; CGE, cyanidin-3-glucoside equivalents; AsA, ascorbic acid; DAsA, dehydroascorbic acid.

**Table 3 plants-10-02521-t003:** Lipid content (mg∙g^−1^ fw) and profiles of fatty acids of the kernels isolated from ripe fruits of different pomegranate cultivars grown in an open field in the south of Italy. Values represent the mean ± standard deviation of three independent sampling replicates (*n* = 3). Data were submitted to one-way analysis of variance (ANOVA) and differences among cultivars, within each category, were detected using multiple comparison procedures (Holm–Sidak post hoc test, *p* < 0.05) and indicated with different letters.

	Pomegranate Cultivars			
	Ako	Emek	Kamel	Wonderful One
Total lipids (mg∙g^−1^ fw)	255 ± 18 ^a^	169 ± 12 ^b^	175 ± 15 ^b^	179 ± 15 ^b^
	**% of total identified fatty acids**			
Palmitic acid (C16:0)	3.26 ± 0.03 ^d^	4.14 ± 0.02 ^b^	4.02 ± 0.03 ^c^	4.57 ± 0.05 ^a^
Stearic acid (C18:0)	3.48 ± 0.02 ^b^	4.15 ± 0.01 ^a^	3.15 ± 0.02 ^c^	4.15 ± 0.02 ^a^
Oleic acid (C18:1 *n*-9)	3.90 ± 0.04 ^d^	4.28 ± 0.03 ^b^	4.06 ± 0.03 ^c^	4.57 ± 0.03 ^a^
Linoleic acid (C18:2 *n*-6)	4.64 ± 0.05 ^c^	5.08 ± 0.04 ^b^	5.27 ± 0.04 ^a^	5.12 ± 0.02 ^b^
Punicic acid (C18:3 *n*-5)	77.65 ± 3.01 ^a^	70.23 ± 2.78 ^b^	75.97 ± 2.10 ^ab^	73.95 ± 2.34 ^ab^
Other C18:3 *n*-3 isomers	7.07 ± 0.07 ^b^	12.12 ± 0.90 ^a^	7.53 ± 0.05 ^b^	7.64 ± 0.06 ^b^
SFA	6.74 ± 0.05 ^d^	8.29 ± 0.03 ^b^	7.17 ± 0.05 ^c^	8.72 ± 0.07 ^a^
MUFA	3.90 ± 0.04 ^d^	4.28 ± 0.03 ^b^	4.06 ± 0.03 ^c^	4.57 ± 0.03 ^a^
PUFA	89.36 ± 3.13 ^a^	87.43 ± 3.72 ^a^	88.77 ± 2.19 ^a^	86.71 ± 2.42 ^a^

SFA, saturated fatty acids; MUFA, monounsaturated fatty acids; PUFA, polyunsaturated fatty acids.

**Table 4 plants-10-02521-t004:** Hydrophilic (HAA), lipophilic (LAA), and total (TAA) antioxidant activities of the juices, kernels, and peels isolated from the ripe fruits of different pomegranate cultivars grown in an open field in the south of Italy. Values represent the mean ± standard deviation of three independent sampling replicates (*n* = 3). Data were submitted to one-way analysis of variance (ANOVA) and differences among groups were detected using multiple comparison procedures (Holm–Sidak post hoc test, *p* < 0.05). For each trait, values within the column followed by different lowercase superscript letters denote significant differences among cultivars, while values within the rows followed by different uppercase superscript letters denote significant differences among fruit parts.

	Fruit Parts		
	Juice	Kernels	Peels
**HAA** **(µmol TE∙g^−1^ fw)**			
Ako	66.0 ± 6.3 ^aB^	29.3 ± 2.9 ^bcC^	368.8 ± 19.0 ^aA^
Emek	68.1 ± 4.0 ^aB^	33.9 ± 5.7 ^bC^	300.7 ± 10.6 ^bA^
Kamel	77.3 ± 7.9 ^aB^	45.2 ± 1.8 ^aC^	294.0 ± 8.9 ^bA^
Wonderful One	51.2 ± 2.1 ^bB^	23.3 ± 1.7 ^cC^	355.5 ± 9.2 ^aA^
**LAA (µmol TE∙g^−1^ fw)**			
Ako	0.2 ± 0.1 ^cC^	1.7 ± 0.1 ^cB^	43.9 ± 1.6 ^dA^
Emek	0.7 ± 0.1 ^bC^	2.4 ± 0.1 ^bB^	74.8 ± 2.5 ^aA^
Kamel	1.4 ± 0.3 ^aC^	4.1 ± 0.3 ^aB^	51.8 ± 3.8 ^cA^
Wonderful One	0.7 ± 0.1 ^bC^	2.2 ± 0.3 ^bcB^	59.3 ± 4.1 ^bA^
**TAA (µmol TE∙g^−1^ fw)**			
Ako	66.2 ± 6.4 ^abB^	31.0 ± 3.0 ^bcC^	412.7 ± 20.6 ^aA^
Emek	68.8 ± 4.1 ^aB^	36.3 ± 5.8 ^bC^	375.5 ± 13.1 ^abA^
Kamel	78.7 ± 8.2 ^aB^	49.3 ± 2.1 ^aC^	345.8 ± 12.7 ^bA^
Wonderful One	51.9 ± 2.2 ^bB^	25.5 ± 2.0 ^cC^	414.8 ± 13.3 ^aA^

TE, Trolox equivalents.

**Table 5 plants-10-02521-t005:** Pearson’s correlation for antioxidant activities (TEAC method) versus antioxidant compounds. *n* (sample size) = 12. Values in bold are significant.

Antioxidants	HAA		LAA		TAA	
	r	*p*	r	*p*	r	*p*
**TSPC**	**0.979**	<0.001	**0.935**	<0.001	**0.981**	<0.001
**TIBPC**	−0.539	0.174	−0.420	0.081	−0.524	0.100
**TFC**	0.318	0.313	0.382	0.221	0.332	0.291
**TA**	−0.231	0.471	−0.260	0.414	−0.238	0.443
**AsA**	**0.982**	<0.001	**0.869**	<0.001	**0.972**	<0.001
**DAsA**	**0.786**	<0.001	**0.708**	0.010	**0.780**	<0.001

HAA, hydrophilic antioxidant activity; LAA, lipophilic antioxidant activity; TAA, total antioxidant activity; r, Pearson’s correlation coefficient; *p*, *p*-value; TSPC, total soluble phenolic compounds; TIBPC, total insoluble-bound phenolic compounds, TFC, total flavonoid compounds, TA, total anthocyanins, AsA, ascorbic acid; DAsA, dehydroascorbic acid.

**Table 6 plants-10-02521-t006:** Concentrations of some elements in the juices, kernels, and peels isolated from ripe fruits of different pomegranate cultivars grown in an open field in the south of Italy. Values (µg∙g^−1^ fw) represent the mean of three independent sampling replicates (*n* = 3). Standard deviation was below 14.5% for all measurements. Data were submitted to one-way analysis of variance (ANOVA) and differences among groups were detected using multiple comparison procedures (Holm–Sidak post hoc test, *p* < 0.05). For each element, values within the row followed by different lowercase superscript letters denote significant differences among cultivars within each fruit part, while different uppercase superscript letters denote significant differences among fruit parts within each cultivar.

		Pomegranate Cultivars—Fruit Parts											
		Ako			Emek			Kamel			Wonderful One		
		Juice	Kernels	Peels	Juice	Kernels	Peels	Juice	Kernels	Peels	Juice	Kernels	Peels
**Element concentration (µg g−1 fw)**	**Ag**	<0.001	<0.001	0.042 ^a^	<0.001	<0.001	<0.001	<0.001	<0.001	0.030 ^b^	<0.001	<0.001	<0.001
	**Al**	0.13 ^bC^	3.04 ^aA^	1.34 ^cB^	0.02 ^aC^	1.93 ^bA^	1.11 ^cB^	0.13 ^bC^	2.34 ^bB^	5.76 ^aA^	<0.001	3.43 ^aB^	4.07 ^bA^
	**As**	0.06 ^bB^	0.22 ^aA^	0.05 ^aB^	0.08 ^aA^	0.10 ^bA^	<0.001	0.07 ^abB^	0.19 ^bA^	0.04 ^aC^	0.07 ^abB^	0.11 ^bA^	0.04 ^aC^
	**B**	2.87 ^aB^	9.61 ^aA^	3.33 ^aB^	2.66 ^bB^	9.10 ^aA^	2.35 ^dB^	2.93 ^aB^	7.59 ^bA^	2.60 ^cC^	2.80 ^abC^	7.40 ^bA^	2.95 ^bB^
	**Ba**	0.13 ^aC^	0.87 ^aB^	1.29 ^abA^	0.04 ^bC^	0.66 ^bA^	0.53 ^cB^	0.04 ^bC^	0.73 ^bB^	1.16 ^bA^	0.04 ^bC^	0.65 ^bB^	1.44 ^aA^
	**Bi**	<0.001	<0.001	<0.001	<0.001	<0.001	<0.001	<0.001	<0.001	<0.001	<0.001	<0.001	<0.001
	**Cd**	<0.001	<0.001	<0.001	<0.001	<0.001	<0.001	<0.001	<0.001	<0.001	<0.001	<0.001	<0.001
	**Co**	<0.001	<0.001	<0.001	<0.001	<0.001	<0.001	<0.001	<0.001	<0.001	<0.001	<0.001	<0.001
	**Cr**	0.04 ^bB^	0.15 ^cA^	0.04 ^bB^	0.05 ^abB^	0.61 ^aA^	0.52 ^aA^	0.06 ^aC^	0.15 ^cA^	0.10 ^bB^	0.05 ^abB^	0.49 ^bA^	<0.001
	**Cu**	0.55 ^bC^	16.94 ^bA^	3.76 ^bB^	0.66 ^aC^	19.08 ^aA^	2.53 ^bcB^	0.65 ^aC^	14.86 ^cA^	7.91 ^aB^	0.63 ^aC^	16.31 ^bA^	1.75 ^cB^
	**Fe**	0.55 ^bC^	71.40 ^aA^	1.61 ^cB^	0.10 ^cC^	74.86 ^aA^	3.94 ^abB^	0.72 ^aC^	49.98 ^bA^	5.61 ^aB^	0.09 ^cC^	65.84 ^aA^	2.95 ^bcB^
	**In**	0.10 ^bB^	0.43 ^aA^	0.38 ^aA^	0.12 ^aC^	0.22 ^cB^	0.36 ^abA^	0.12 ^aC^	0.34 ^bB^	0.40 ^aA^	0.11 ^abC^	0.15 ^dB^	0.30 ^bA^
	**K**	2152 ^bB^	3440 ^bA^	4103 ^aA^	2121 ^bC^	4900 ^aA^	3430 ^aB^	2629 ^aB^	3914 ^bA^	4124 ^aA^	2516 ^aB^	3782 ^bA^	4010 ^aA^
	**Li**	<0.001	<0.001	<0.001	<0.001	<0.001	<0.001	<0.001	<0.001	<0.001	<0.001	<0.001	<0.001
	**Mg**	50 ^bC^	1151 ^bA^	249 ^aB^	57 ^aC^	1420 ^aA^	150 ^bB^	60 ^aC^	980 ^bA^	286 ^aB^	58 ^aC^	953 ^bA^	235 ^aB^
	**Mn**	0.41 ^bC^	16.60 ^aA^	1.91 ^aB^	0.45 ^aC^	15.87 ^aA^	1.44 ^cB^	0.42 ^abC^	12.22 ^bA^	1.68 ^bB^	0.42 ^abC^	12.29 ^bA^	1.90 ^aB^
	**Mo**	<0.001	0.16 ^b^	<0.001	<0.001	0.16 ^b^	<0.001	<0.001	0.23 ^a^	<0.001	<0.001	0.24 ^a^	<0.001
	**Na**	16 ^bC^	283 ^aA^	81 ^bB^	17 ^bC^	213 ^bA^	41 ^dB^	14 ^bC^	207 ^bA^	71 ^cB^	24 ^aC^	165 ^bA^	186 ^aB^
	**Ni**	0.09 ^cC^	0.19 ^bA^	0.13 ^bB^	0.14 ^aC^	0.27 ^aB^	0.40 ^aA^	0.12 ^bB^	0.21 ^bAB^	0.41 ^aA^	0.06 ^dC^	0.28 ^aA^	0.09 ^bB^
	**Pb**	0.04 ^bB^	0.21 ^aA^	0.03 ^cB^	0.07 ^aB^	0.10 ^bA^	0.07 ^bB^	0.07 ^aB^	0.19 ^aA^	0.14 ^aAB^	0.07 ^aB^	0.19 ^aA^	0.05 ^bcB^
	**Sr**	0.04 ^bC^	3.21 ^bB^	3.75 ^bA^	0.06 ^aC^	3.55 ^aA^	1.89 ^cB^	0.07 ^aC^	2.98 ^cB^	3.93 ^bA^	0.04 ^bC^	2.72 ^dB^	4.44 ^aA^
	**Te**	0.07 ^aA^	0.10 ^bA^	0.21 ^aA^	0.07 ^aC^	0.31 ^aA^	0.18 ^aB^	0.07 ^aA^	0.11 ^bA^	0.12 ^bA^	0.06 ^aB^	0.33 ^aA^	0.08 ^bB^
	**Tl**	<0.001	<0.001	<0.001	<0.001	<0.001	<0.001	<0.001	<0.001	<0.001	<0.001	<0.001	<0.001
	**V**	0.06 ^bC^	2.64 ^aA^	0.23 ^bB^	0.08 ^aB^	2.80 ^aA^	0.10 ^dB^	0.08 ^aC^	1.79 ^bA^	0.26 ^aB^	0.08 ^aC^	1.69 ^bA^	0.20 ^cB^
	**Zn**	0.82 ^cC^	36.31 ^aA^	1.35 ^bB^	1.00 ^bC^	34.62 ^abA^	2.16 ^bB^	0.97 ^bC^	27.74 ^cA^	5.30 ^aB^	1.34 ^aC^	32.85 ^bA^	2.03 ^bB^

**Table 7 plants-10-02521-t007:** Variables participating in the construction of the factorial axes and their relative contribution (%) to PCA dimensions.

Variables	Contribution (%)	
	PC1	PC2
Moisture	6.291	0.035
TESC	4.566	0.878
TEIC	6.013	0.084
TSPC	1.150	9.006
TIBPC	6.272	0.077
TFC	2.730	5.945
TA	0.011	3.722
AsA	1.352	8.246
DAsA	1.157	4.648
HAA	1.453	8.591
LAA	0.806	8.740
TAA	1.357	8.778
Al	1.122	4.660
As	3.816	1.524
B	6.106	0.163
Ba	0.437	8.814
Cr	2.046	0.235
Cu	6.226	0.066
Fe	6.261	0.074
In	0.587	6.478
K	1.663	6.898
Mg	6.213	0.017
Mn	6.286	0.018
Na	4.907	0.639
Ni	0.696	2.705
Pb	4.344	0.074
Sr	1.604	7.605
Te	2.101	1.154
V	6.093	0.050
Zn	6.334	0.075

TESC, total ethanol-soluble carbohydrates; TEIC, total ethanol-insoluble carbohydrates; TSPC, total soluble phenolic compounds; TIBPC, total insoluble-bound phenolic compounds; TFC, total flavonoid compounds; TA, total anthocyanins; AsA, ascorbic acid; DAsA, dehydroascorbic acid; HAA, hydrophilic antioxidant activity; LAA, lipophilic antioxidant activity; TAA, total antioxidant activity.

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
