# Peer review of "Analysis of the Phytochemical Composition of Pomegranate Fruit Juices, Peels and Kernels: A Comparative Study on Four Cultivars Grown in Southern Italy"

_plants, 2021, doi:10.3390/plants10112521_

Round 1
Reviewer 1 Report
The article is of interest and is well written; it would only be desirable for it to be complete, to provide the determinations not made in Table 1, or, failing that, to justify why it has not carried out those determinations.
Author Response
Referee 1
The article is of interest and is well written; it would only be desirable for it to be complete, to provide the determinations not made in Table 1, or, failing that, to justify why it has not carried out those determinations.
We thanks the referee for the excellent evaluation of our manuscript.
With regard to the suggestion of completing table 1, we reputed pH and TSS determinations unappropriate for kernel characterization. Unfortunately, at the moment of the research (2016-2017), determination of organic acids in the kernels was not planned. Although, although we recognize that this is a shortcoming, currently we have no more samples of the same batch to perform the analysis.

Reviewer 2 Report
The current work, Fractionate analysis of the phytochemical composition of pomegranate fruits: a comparative study on four cultivars grown in Southern Italy fits within the scope of the journal Plants and results can be considered of interest for breeders and growers in order to cultivate the pomegranate fruits with high nutritional value.
The manuscript presents extensive experimental material, however, I cannot understand the scientific significance of this work. This manuscript just seems to list the results obtained by from a large number of chemical analyses. I cannot accept this type of just a collection of results as a scientific report. The proper interpretation and discussion must be necessary.
In present version of manuscript, most of the discussion is devoted to comparing the data obtained on the content of certain components in pomegranate fruits with the available data in the literature. However, as the authors themselves note, there can be many reasons for the differences (different growing conditions, genotypes, cultivation practices, tree age and extraction methods), and therefore it is incorrect to make a comparison.
I encourage the authors to discuss also why such results were obtained for different phytochemicals contents in various parts of the pomegranate fruits. For example, why is the content of soluble phenolics higher in the peels, and not in the juice? Why the highest content of flavonoids was detected in kernels? Where does the biosynthesis of these compounds take place? How and where does the transport of these compounds take place in pomegranate plants? Etc.
In Abstract, Results and Conclusion Sections authors write: “The levels of phenols, flavonoids, anthocyanins, ascorbic acid and dehydroascorbic acid, as well as the antioxidant activity of all samples were higher than literature reported data, likely due to positive interactions among genotype, environment and good agricultural practices”. If authors make this conclusion, I strongly recommend add to Materials and Methods Section the information about the pedoclimatic characteristics of the Salento region as well as about “agricultural practices” that were used in the cultivation of pomegranates in this experiment.
For better understanding of obtained data it is very important to make a principal component analysis or heat-map analysis with clustering. This will show a good relation between your variable and difference between studied cultivars and fruit parts. I think it’s very necessary for this kind of data.
Specific suggestions and comments are provided below:
1. Introduction: Please avoid writing long paragraphs, try to shorten all long paras so that confusion is avoided (this remark also applies to the results section ). And just write 2-3 paras in Introduction section. Some information in Introduction (for example L73-82) can be reduced.
2. L 448: Please check. It should be rows, not column. Why authors check significance of difference among fractions only for elements? and not for other studied parameters?
3. What authors mean under three independent replicate experiments? Was it only technical/analytical replications? Or authors did repeat whole experiment three times beginning from harvesting of fruits?
4. Finally, I think they are way to many references for a paper like this, they should be reduce up to 60 if its possible with in the last 10 years.
Author Response
Referee 2
The current work, Fractionate analysis of the phytochemical composition of pomegranate fruits: a comparative study on four cultivars grown in Southern Italy fits within the scope of the journal Plants and results can be considered of interest for breeders and growers in order to cultivate the pomegranate fruits with high nutritional value.
The manuscript presents extensive experimental material, however, I cannot understand the scientific significance of this work. This manuscript just seems to list the results obtained by from a large number of chemical analyses. I cannot accept this type of just a collection of results as a scientific report. The proper interpretation and discussion must be necessary.
In present version of manuscript, most of the discussion is devoted to comparing the data obtained on the content of certain components in pomegranate fruits with the available data in the literature. However, as the authors themselves note, there can be many reasons for the differences (different growing conditions, genotypes, cultivation practices, tree age and extraction methods), and therefore it is incorrect to make a comparison.
I encourage the authors to discuss also why such results were obtained for different phytochemicals contents in various parts of the pomegranate fruits. For example, why is the content of soluble phenolics higher in the peels, and not in the juice? Why the highest content of flavonoids was detected in kernels? Where does the biosynthesis of these compounds take place? How and where does the transport of these compounds take place in pomegranate plants? Etc.
We have discussed the results according to reviewer suggestion adding the following sentences:
Pg. 11-12, lines 342-359 of the R1 version in track change mode: “With regard to the three fractions, the differences observed in the accumulation of phenolic and flavonoids are essentially due to fruit tissue specialization. Peels accumulate preferentially phenolic compounds acting as protective screens against high-energy wavelengths to reduce light-induced oxidative stress and fruit sunburn, as well as those with toxic effects or protein precipitation capacity as a line of defense against pathogens and predators. The cells of the juicy sarcotesta preferentially synthesize anthocyanins at the cytosolic surface of the endoplasmic reticulum and transport them across tonoplast through specific carriers (ATP-binding cassette transporter and multidrug and toxic compound extrusion proteins) within large vacuoles. Vacuolar anthocyanins contribute to the red bright, brilliant color of arils fundamental to attract seed dispersers. Kernels are characterized, instead, by high levels of insoluble-bound phenols likely involved in hardening the thick wall of sclerotesta cells. Phenols (both soluble and insoluble) play, also, vital roles in the protection of the zygotic and reserve tissues from biotic aggression and abiotic stresses favoring seed survival, as well as in seed development, maturation, dormancy and germination. An increase in the ratio of insoluble-bound to soluble phenolic compounds was observed in lentil seeds during germination, suggesting that phenol secretion to the cell wall followed by insolubilization may exert a pivotal action in regulating the process [56,57].”,
We also added the following related references:
“56. Corso, M.; Perreau, F.; Mouille, G.; Lepiniec, L. Specialized phenolic compounds in seeds: structures, functions, and regulations. Plant Sci. 2020, 296, 110471. doi:10.1016/j.plantsci.2020.110471
57. Shahidi, F.; Yeo, J.D. Insoluble-bound phenolics in food. Molecules 2016, 21(9), 1216. doi:10.3390/molecules21091216”
In Abstract, Results and Conclusion Sections authors write: “The levels of phenols, flavonoids, anthocyanins, ascorbic acid and dehydroascorbic acid, as well as the antioxidant activity of all samples were higher than literature reported data, likely due to positive interactions among genotype, environment and good agricultural practices”. If authors make this conclusion, I strongly recommend add to Materials and Methods Section the information about the pedoclimatic characteristics of the Salento region as well as about “agricultural practices” that were used in the cultivation of pomegranates in this experiment.
We added the required information in the Materials and Methods section, pg. 19, lines 572-589:
“The average annual climate parameters of the experimental region are: temperature 17°C (with a maximum of 30.2°C in August and a minimum of 9.9°C in January), precipi-tation 606-640 mm, relative humidity 74-61%, average UV index 7-4. The soil is a red earth dark in color moderately to very deep with little structure and a clay-loam texture classi-fied as Alfisol according to the US Soil Taxonomy.
The plants, aged 5 years, were arranged in the field at a distance of 6 m between the rows and 2.5 m in the row and supported by galvanized iron Y-shaped poles holding two lines of steel wires for arm. Mulching was made on the row with white plastic film (thick-ness 100 mm, width 1.5 m), tucked up 25-30 cm on both sides and with about 1 m of free span to reduce weeds and water consumption, and to increase the refraction of sunlight towards the canopy. A complete fertigation system, with a dripline of 16 mm per row with drips at 50 cm and a capacity of 2.2 L‧h-1, was used with an average total water consump-tion for year of 3500 m3‧ha-1. Fertigation was applied with nitrogen, phosphorous and po-tassium (250, 50-60 and 250 Units‧ha-1 per year, respectively, split for the entire production cycle) and chelated iron (3 applications per year of 50 kg‧ha-1 each). Brown pruning was carried out in dormant season (January), green pruning was carried out after fruit set (July) for sucker removal and thinning. Aphid control was carried out using diluted liquid soap or white oil, when required.”
For better understanding of obtained data it is very important to make a principal component analysis or heat-map analysis with clustering. This will show a good relation between your variable and difference between studied cultivars and fruit parts. I think it’s very necessary for this kind of data.
We performed the PCA analysis as required by the referee, for this reason an additional figure (Figure 5) and an additional table (Table 7), and their relative captions were added. The results were discussed in view of contributing to explain why differences in the contents of different phytochemical were obtained in the various fractions of pomegranate fruits as follows (Pag. 16-17, lines 499-551):
“2.4. Principal component analysis
To highlight biochemical differences among fractions and groups of genotypes and to highlight any clustering of the observations, a multivariate analysis (PCA) was carried out (Figure 5). Two relevant principal components (PCs) that explain up to 81.57% of the variation of the collected data were extracted. The first axis (PC1) explains 52.09% of the total variance, while the second (PC2) the remaining 29.48%. The contribution of each quality parameter (variables) is reported in Table 7.
PC1 vs. PC2 plot show clear clustering of fruit fractions. Irrespectively of cultivars, all juices grouped on the lower-left side of the chart showing high moisture, and high con-centrations of soluble carbohydrates and anthocyanins. Anthocyanin level in the pome-granate fruit is developmentally regulated. Yuan, et al. [79] identified 26 anthocyanin bio-synthesis candidate genes, which showed tissue-specific expression in arils and peels. Indeed, high anthocyanin levels in both fractions are a key factor for varietal selection be-cause largely determine the economic value of the fruit.
Kernels were instead distributed in the mid-right side of the chart being characterized by high levels of metallic minerals, ethanol-insoluble carbohydrates and insoluble-bound phenolics. The close correlation between these variables confirms that insoluble phenols are mainly bound to structural polysaccharides, which in turn play a role in accumulat-ing/sequestering metal ions contributing to cell wall architecture or those potentially toxic for the protoplast. Boron, for example, is an important structural element of plant cell wall as it is involved in rhamnogalacturonan‐II crosslinking.
Peels were grouped in the upper left pane showing large amounts of soluble phenols, AsA and DAsA, as well as high antioxidant activities. This possibly reflects the protective role that peel phenolic compounds exert against solar UV radiations and/or pathogen infections.”
Specific suggestions and comments are provided below:
- Introduction: Please avoid writing long paragraphs, try to shorten all long paras so that confusion is avoided (this remark also applies to the results section). And just write 2-3 paras in Introduction section. Some information in Introduction (for example L73-82) can be reduced.
We have shortened the introduction by removing unnecessary information according to the suggestion of the referee. When possible we also tried to shorten long sentences in the results section to make them more readable.
- L 448: Please check. It should be rows, not column. Why authors check significance of difference among fractions only for elements? and not for other studied parameters?
Right, the proper term was row, not column. We have corrected the sentence.
Significance for differences among fractions was added for all studied parameters. Table and figure legends were modified accordingly.
- What authors mean under three independent replicate experiments? Was it only technical/analytical replications? Or authors did repeat whole experiment three times beginning from harvesting of fruits?
We repeated the whole experimet three times starting from different batches of fruits. This was better explained by modifing lines 707-709 as follows: “Results are presented as the mean value ± standard deviation of three independent biological replicate experiments (n = 3; for each cultivar three different batches of fruits (at least 3 kg from the same harvesting lot) were separately processed and analyzed.”
- Finally, I think they are way to many references for a paper like this, they should be reduce up to 60 if its possible with in the last 10 years.
The number of references has bee significantly reduced.

Round 2
Reviewer 2 Report
The authors improved the manuscript well. I am satisfied with most of the authors' responses to comments, but I still have one remark:
Please add to the titles of revised Tables 1, 2, 4 the information about lowercase and uppercase letters similar to Table 6. As an example: “For each trait, values within the column followed by different lowercase superscript letters denote significant differences among cultivars, while values within the rows followed by different uppercase superscript letters denote significant differences among fractions”.
Author Response
Please add to the titles of revised Tables 1, 2, 4 the information about lowercase and uppercase letters similar to Table 6. As an example: “For each trait, values within the column followed by different lowercase superscript letters denote significant differences among cultivars, while values within the rows followed by different uppercase superscript letters denote significant differences among fractions”.
We added the required information to Tables 1, 2, 4, according to the reviewer observation.
We sincerely wish to thank the reviewer for the excellent suggestions made during the first and second rounds of review, which helped us to significantly improve the work and gave us insight into other subsequent work.